# The Combination of Sleep Disorders and Depression Significantly Increases Cancer Risk: A Nationwide Large-Scale Population-Based Study

**DOI:** 10.3390/ijerph19159266

**Published:** 2022-07-28

**Authors:** Fang-Chin Hsu, Chih-Hsiung Hsu, Chi-Hsiang Chung, Ta-Wei Pu, Pi-Kai Chang, Tzu-Chiao Lin, Shu-Wen Jao, Chao-Yang Chen, Wu-Chien Chien, Je-Ming Hu

**Affiliations:** 1Department of Surgery, Tri-Service General Hospital, National Defense Medical Center, Taipei 114, Taiwan; gra231934@gmail.com; 2Graduate Institute of Medical Sciences, National Defense Medical Center, Taipei 114, Taiwan; chihhsung@gmail.com; 3Teaching Office, Tri-Service General Hospital, National Defense Medical Center, Taipei 114, Taiwan; 4School of Public Health, National Defense Medical Center, Taipei 114, Taiwan; g694810042@gmail.com; 5Department of Medical Research, Tri-Service General Hospital, National Defense Medical Center, Taipei 114, Taiwan; 6Taiwanese Injury Prevention and Safety Promotion Association, Taipei 114, Taiwan; 7Division of Colon and Rectal Surgery, Department of Surgery, Songshan Branch, Tri-Service General Hospital, National Defense Medical Center, Taipei 114, Taiwan; tawei0131@gmail.com; 8Division of Colorectal Surgery, Department of Surgery, Tri-Service General Hospital, National Defense Medical Center, Taipei 114, Taiwan; pencil8850@hotmail.com (P.-K.C.); joelin814@gmail.com (T.-C.L.); jaosw@ndmctsgh.edu.tw (S.-W.J.); cartilage77@yahoo.com.tw (C.-Y.C.); 9Graduate Institute of Life Sciences, National Defense Medical Center, Taipei 114, Taiwan; 10School of Medicine, National Defense Medical Center, Taipei 114, Taiwan

**Keywords:** sleep disorder, depression, cancer

## Abstract

**Introduction:** Sleep disorders, depression, and cancer have become increasingly prevalent worldwide. However, it is unknown whether coexistence of sleep disorders and depression influences the risk of cancer development. Therefore, we conducted a nationwide population-based study to examine this association among patients in Taiwan. **Materials and Methods:** A total of 105,071 individuals diagnosed with cancer and 420,284 age- and sex-matched patients without a diagnosis of cancer between 2000 and 2015 were identified from Taiwan’s National Health Insurance Research Database. **The underlying chronic diseases of patients that may developed cancer were gathered and studied as the predictor**. A multivariate Cox proportional odds model was used to estimate the crude and adjusted odds ratios (aORs) with 95% confidence intervals (CIs) to estimate the interaction effect between sleep disorders and depression on the risk of cancer. **Results:** After adjusting for age, sex, comorbidities, and other covariates, the cancer group was associated with increased exposure to sleep disorders than the non-cancer group (aOR = 1.440, 95% CI = 1.392–1.489, *p* < 0.001). In addition, patients with both sleep disorders and depression were at an even higher risk for cancer than the general population (aOR = 6.857, *p* < 0.001). **Conclusions:** This retrospective cohort study shows that patients with both sleep disorders and depression are at a higher risk of cancer. **Clinically, a meticulous cancer risk evaluation is recommended for patients with both sleep disorders and depression.**

## 1. Introduction

Sleep is a naturally recurring state of the mind and body that accounts for approximately one-third of a person’s life. Sleep enables the human body to recover after waking activities, ensuring optimal subsequent functioning [1]. However, the prevalence of sleep disorders has increased worldwide, ranging from approximately 10% to 30% in Western countries [2,3] and approximately 20% to 30% in Asian populations [4,5]. According to the Diagnostic and Statistical Manual of Mental Disorders, Fifth Edition (DSM-V), people with sleep disorders are dissatisfied with their quality, timing, and amount of sleep, resulting in daytime distress and impairment. Numerous detrimental effects have been reported, including the risk of cardiovascular disease, diabetes mellitus, obesity, and mortality [6,7,8].

In patients with cancer, sleep disorders often occur because of the physical or psychological impact of the cancer or its treatment. However, several studies have shown that sleep disorders may be associated with the development of cancers, such as breast cancer [9], colorectal cancer [10], lung cancer [11,12], and liver cancer [12]. Furthermore, sleep disorders are a serious health challenge and often coexist with mental disorders, such as major depression disorder (MDD) [13].

Since the 1980s, research has shown that the immune system, the endocrine system, cancer metastasis, treatment tolerance, and other processes are all impacted by depression [14]. It has been suggested that depression and anxiety, the two most prevalent mental disorders with past-year prevalence rates of 5% and 7%, respectively, in the general population, may potentially play an etiologic role and have a prognostic impact on cardiovascular diseases, such as stroke and coronary heart disease [15,16]. Furthermore, depression is much more prevalent among cancer patients, affecting up to 20% of cancer patients [17]. However, when investigating the relationship between depression and cancer, most researchers have focused on cancer-induced depression, and only few have explored depression as a risk factor for subsequent cancer. Although some meta-analysis studies have investigated the relationship between depression and overall cancer risk over the past decade [14,18], the results of these studies are inconsistent or require larger sample sizes for confirmation. 

To the best of our knowledge, no study has examined if and how the coexistence of sleep disorders and depression impacts cancer risk in the human population. Thus, we conducted a population-based cohort study to assess the interaction effect between sleep disorders and depression on cancer risk in a large general population in Taiwan using a longitudinal health insurance dataset selected from the National Health Insurance Research Database (NHIRD).

## 2. Materials and Methods

### 2.1. Data Source

The current study was a population-based retrospective cohort study conducted using the medical claims dataset from Taiwan’s NHIRD. The National Health Insurance (NHI) program is a single-payer mandatory enrollment program implemented by the government of Taiwan in 1995, enrolling up to 99% of Taiwan’s population (approximately 23 million people) [19]. The NHIRD contains comprehensive, high-quality information on epidemiological research and data on diagnoses, prescription use, and health-care information, including inpatient/ambulatory claims, prescription claims, demographic data, and hospitalization; these data have been widely used in academic studies [20,21]. Diseases in the database were defined using International Classification of Disease, Ninth Revision (ICD-9) codes.

### 2.2. Study Subjects

Patients with cancer between January 2000 and December 2015 were selected from the longitudinal health insurance dataset and categorized according to ICD codes (140–149, 150, 151, 153–154, 155, 160–161, 162, 174, 179–184, 185, 88–189, 193, 200–208); this group comprised the study cohort (Figure 1). To identify patients with sufficient accuracy, we ascertained patients with cancer according to the Registry for Catastrophic Illness Patient Database (RCIPD), a subset of the NHIRD. In Taiwan, a patient must provide an official hospital diagnosis certificate with confirmation based on laboratory results, pathology, and/or diagnostic imaging in order to obtain a certificate of catastrophic disease. Therefore, the diagnosis of cancer in our study was reliable. We also enrolled a 1:4 age-, sex-, and index year-matched group of patients from the NHIRD. These patients were enrolled as the unexposed group or control cohort since they were not diagnosed with cancer throughout the study period. The diagnoses of all sleep disorders were made by certified psychiatrists and in accordance with the DSM-V criteria, which required at least two outpatient visits or one admission record for ICD-9-CM codes: 307.4 and 780.5. Patients under the age of 18, those of unknown sex, those who used drugs, and those who had the last sleep disorders (SD) diagnosis before the first cancers diseases diagnosis, but less than 2 years in retrospective duration, were all excluded.

### 2.3. Variables of Interest

The covariates included age, sex, monthly insured premiums, comorbidities, locations, urbanization levels of residence, and the level of healthcare provided at the facility of diagnosis. Covariates that were potential confounders in the association between sleep disorders and cancer included age, sex, and underlying chronic diseases. The chronic diseases of patients that may developed cancer were gathered and studied as the predictor, and they included hypertension (HTN, ICD-9-CM codes: 401, 404, and 405), diabetes mellitus (DM) (ICD-9-CM code: 250), stroke (ICD-9-CM code: 430–438), dementia (ICD-9-CM codes: 290, 294.1, and 331.0), chronic kidney disease (CKD, ICD-9-CM code: 585), and depression. All depression diagnoses were made by certified psychiatrists and in accordance with the DSM-V criteria, which required at least two outpatient visits or one admission record for ICD-9-CM codes: 296.2–296.3, 296.82, 300.4, 311.

### 2.4. Statistical Analysis

The distribution of comorbidities and sociodemographic data was compared between the case group (with cancer) and the control group (without cancer) using the chi-squared test to examine categorical variables and Student’s *t*-test to examine continuous variables. We computed the odds ratios (ORs) and 95% confidence intervals (CIs) using logistic proportional odds models after adjusting for the potential confounders mentioned above (including age, sex, monthly insured premiums, comorbidities, locations, urbanization levels of residence, and the level of healthcare provided at the facility of diagnosis). All confounders, such as covariates and comorbidities, were calculated separately. Further analysis was done to assess the interaction effect between sleep disorders and depression on the risk of cancer, and adjusted ORs were also estimated for 12 specific cancer types using conditional logistic regression. All statistical analyses were performed using the SPSS software (version 22.0; SPSS Inc., Chicago, IL, USA). For two-sided testing, the significance level was set at *p* < 0.05.

## 3. Results

In total, 525,355 patients were enrolled in this study, including 105,071 adult patients with cancer and 420,284 patients without cancer. The percentage of pre-existing sleep disorders was 4.92% in the case group and 3.86% in the control group. The participants’ age and sex, monthly insured premium, comorbidities, location and urbanization level of residence, and level of hospitals for medical help are summarized in Table 1. There were no significant differences in age, sex, or insurance premiums (NT$). Most patients were aged ≥65 years. The cancer group tended to have a higher prevalence of HTN, depression, stroke, and CKD. Furthermore, cancer patients tended to live in Taiwan’s north and south, in areas with a high level of urbanization, and sought medical treatment from tertiary hospital centers.

Table 2 shows the results of the logistic regression analysis of the risk factors for cancer. A significantly higher risk of cancer development was observed in patients with sleep disorders. The crude OR was 1.789 (95% CI = 1.648–1.991, *p* < 0.001). After adjusting for age, sex, comorbidities, geographical area of residence, urbanization level of residence, and monthly income, the adjusted OR was 1.440 (95% CI = 1.392–1.489; *p* < 0.001). In addition, male sex, age ≥ 45 years, and presence of DM, HTN, depression, stroke, dementia, and/or CKD were associated with an increased risk of cancer development.

The overall incidence of cancer was 17.94 per 1000 person-years in the study cohort and 6.83 per 1000 person-years in the control cohort in the subgroup analysis comparing patients with and without sleep disturbances (Table 3). Sleep disturbances were linked to an elevated risk of cancer in both female and male patients (1.399 in the corresponding age group of women, *p* < 0.001; 1.469 in the corresponding age group of males, *p* < 0.001). On age group analysis, the presence of sleep disorders in all age groups was independently associated with an increased risk of cancer diagnosis relative to the absence of sleep disorders (aOR = 1.280, *p* < 0.001 in 18–44, aOR = 1.280, *p* < 0.001 in 45–64, aOR = 1.280, *p* < 0.001 in ≥65 years). In the preexisting condition group analysis, patients with sleep disorders showed an increased risk of developing cancer, regardless of whether they had DM, HTN, depression, stroke, dementia, or CKD (aOR = 1.338–2.854, all *p* < 0.001). Interestingly, patients with sleep disorders showed an increased risk of developing cancer, regardless of the season (aHR = 1.302–1.633, *p* < 0.001), urbanization level (aHR = 1.389–1.506, *p* < 0.001), or level of care group analysis (aHR = 1.420–1.820, *p* < 0.001).

Notably, patients with sleep disorders were associated with a higher risk of cancer than those without sleep disorders, and patients with both sleep disorders and depression were associated with an even higher risk of cancer (aOR = 6.857, *p* < 0.001) (Figure 2). In the subgroup analysis of specific cancer types, our data suggest an increased risk of 12 types of cancer, especially esophageal and hematologic cancers, in patients with both sleep disorders and depression (Figure 3).

## 4. Discussion

To the best of our knowledge, this is the first large-scale, population-based study of Asian individuals that investigated the interaction effect between sleep disorders and depression, particularly MDD, on the risk of developing cancer. There is growing evidence that sleep disorders are associated with an increased incidence of cancer [12,22,23]. However, previous studies investigating the relationship between depression and cancer showed inconsistent results [14,24,25]. The most valuable finding of our study is that patients with coexisting sleep disorders and depression were 6.85 times more likely to develop cancer than the general population.

Previous studies in animal models have demonstrated that circadian disruption may increase the risk of cancer development [26,27]. Additionally, intermittent hypoxia and sleep fragmentation mimicking obstructive sleep apnea (OSA) are known to accelerate tumor growth and invasiveness [28,29]. However, evidence of the association between sleep disorders and cancer in humans is more limited than in animal models, and most studies have focused on the association between shift work and cancer [30,31]. Our study showed that patients with sleep disorders had a higher cancer risk than those without sleep disorders. This result is compatible with previous studies in which the researchers also applied data from the NHIRD using different study designs [12,23]. Several mechanisms could explain the link between sleep disorders and cancer incidence. For example, melatonin production is suppressed in sleep disorders. Melatonin can neutralize and remove free radicals and has a protective effect against the DNA-damaging effects of hydrogen peroxide [32]. Decreased melatonin inactivates melatonin receptor MT1 and increases expression of the tumor suppressor gene p53 [33], leading to cancer. Another possible mechanism could be immunosuppression. Sleep and the circadian system are strong regulators of immunological processes. Suppression of the immune system may lead to the establishment and growth of malignant clones [34]. In patients with obstructive sleep apnea, intermittent hypoxia and sleep fragmentation may lead to increased oxidative stress and systemic inflammation, resulting in an increased risk of tumors [22].

Depression and cancer commonly co-occur. Most studies on depression and cancer have focused on diagnosing psychological illnesses in cancer patients to increase the treatment effects on both cancer and depression. Recent epidemiological studies suggested that new-onset depression may be associated with a modestly increased risk of certain cancers [35,36]. Our findings align with these results and contribute to the evidence by showing that patients with both sleep disorders and depression had a high risk of cancer, and this risk was approximately six times higher than that in the general population. Several mechanisms may explain the association between depression and cancer incidence. The first possible mechanism involves dysregulation of the hypothalamic–pituitary–adrenal (HPA) axis, especially diurnal variations in cortisol and melatonin [37]. Burke et al. reported that depression is associated with blunted cortisol stress reactivity and impaired stress recovery in patients with MDD [38]. Alteration in cytokine regulation is another possible explanation for the increase in incident cancer [39]. Howren et al. found a positive association between C-reactive protein (CRP), interleukin-1 (IL-1), interleukin-6 (IL-6), and depression [40]. Elevated brain IL-1 levels are sufficient to produce a high incidence of depression [39], and IL-1 is known to exert a critical function in malignancies, influencing the tumor microenvironment and promoting cancer initiation and progression [41]. Moreover, Nunes et al. conducted a study including 40 non-medicated, depressed adults and 34 healthy non-depressed adults, and found that their immune and hormonal measurements differed significantly [42]. Depression may predispose individuals to an unhealthy lifestyle or behavior that puts them at a higher cancer risk [24]. 

This study recorded a significantly higher risk of hematologic and esophageal cancer in patients with coexisting sleep disorders and depression, but no significant increase was found in the incidence of aggressive breast cancer. By contrast, several studies have suggested that women who are short sleepers may develop more aggressive breast cancers than those who sleep longer hours [9,43,44]. A possible reason for this inconsistency may be related to the Taiwanese free cancer screening program. Cervical, oral, breast, and colorectal cancers were the main targets of the screening program. Early screening can detect carcinoma in situ at an early stage; thus, patients could receive cancer treatment before the disease progresses or becomes invasive. Moreover, epidemiologic studies have shown that patients with sleep disorders frequently practice unhealthy lifestyles, including smoking and drinking alcohol [45,46]. Chronic smoking and alcohol consumption are the primary etiologies of esophageal cancer, especially squamous cell carcinoma, which is the most common type of esophageal cancer in Taiwan. These mechanisms combined with the suppressed immune system due to insufficient sleep may explain the results of the present study.

The case-control study design and detailed information regarding various potential confounders constitute the critical strengths of our study. However, this study had several limitations that must be considered. First, despite controlling for various potential confounding factors, certain demographic variables, such as socioeconomic status and family history of cancer, could not be considered because the corresponding data were not available. Second, we did not obtain information on sleep duration and lifestyle (e.g., smoking and drinking), which can be important additional factors when assessing the severity of sleep disorders. Finally, future studies with more extended follow-up periods are necessary to detect certain cancers.

## 5. Conclusions

This retrospective cohort study showed that patients with coexisting sleep disorders and MDD were 6.85 times more likely to develop cancer than the general population, indicating that the interaction between sleep disorders and depression may synergistically increase cancer occurrence. The results of this study could serve as a reminder for clinicians who provide long-term care for patients with consistent sleep disorders and MDD. Further biological mechanistic research and prospective studies are needed to confirm our findings. Clinically, a meticulous cancer risk evaluation is recommended for patients with both sleep disorders and depression.

## Figures and Tables

**Figure 1 ijerph-19-09266-f001:**
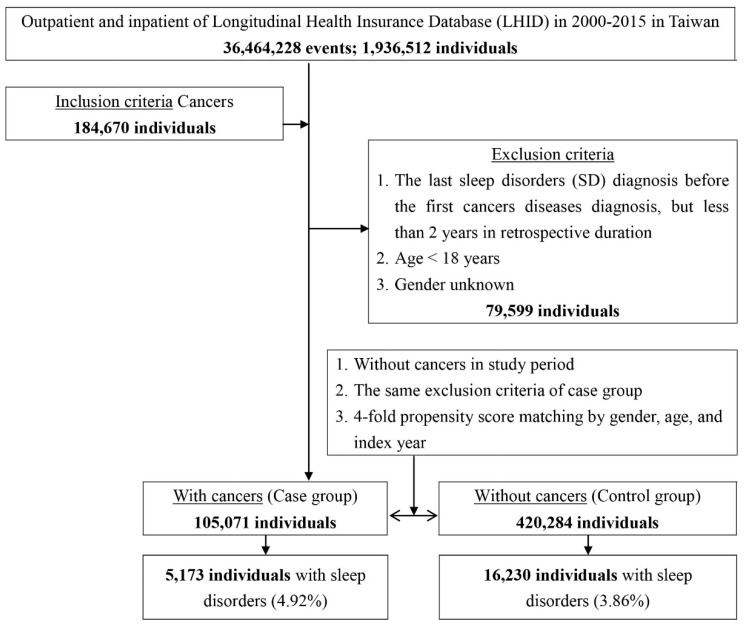
Flowchart of participant selection (nested case-control study) from the National Health Insurance Research Database in Taiwan.

**Figure 2 ijerph-19-09266-f002:**
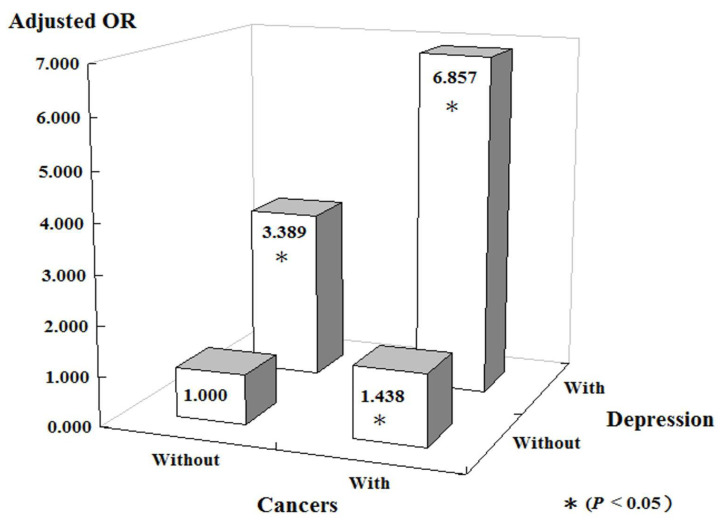
Incidence of cancers stratified by sleep disorders and depression using conditional logistic regression. OR, odds ratio.

**Figure 3 ijerph-19-09266-f003:**
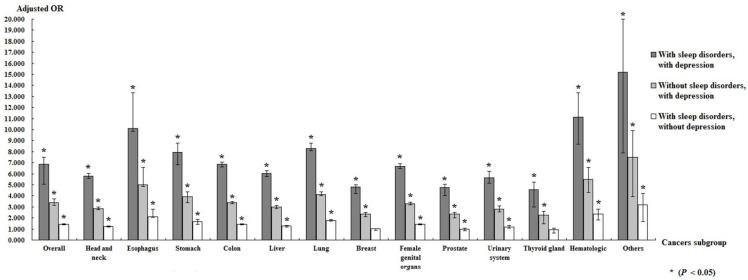
Subgroup analysis of specific cancer types in patients with both sleeping disorders and depression.

**Table 1 ijerph-19-09266-t001:** Characteristics of the study population.

Cancers	Total		With		Without		*p*
Variables	n	%	n	%	n	%	
**Total**	525,355		105,071	20.00	420,284	80.00	
**Gender**							0.999
Male	300,050	57.11	60,010	57.11	240,040	57.11	
Female	225,305	42.89	45,061	42.89	180,244	42.89	
**Age (years)**	63.42 ± 16.63		63.49 ± 15.13		63.40 ± 16.99		0.117
**Age group (years)**							0.999
18–44	68,105	12.96	13,621	12.96	54,484	12.96	
45–64	192,775	36.69	38,555	36.69	154,220	36.69	
≥65	264,475	50.34	52,895	50.34	211,580	50.34	
**Married**							0.004
Without	67,423	12.83	13,445	12.80	53,978	12.84	
With	276,913	52.71	55,697	53.01	221,216	52.63	
Divorce	61,468	11.70	12,456	11.85	49,012	11.66	
Spouse death	119,467	22.74	23,454	22.32	96,013	22.84	
Unknown	84	0.02	19	0.02	65	0.02	
**Education**							<0.001
Elementary/junior high school	50,760	9.66	10,154	9.66	40,606	9.66	
(Vocational) high school	296,352	56.41	59,901	57.01	236,451	56.26	
Univeristy/college/graduate	178,211	33.92	35,010	33.32	143,201	34.07	
Others	32	0.01	6	0.01	26	0.01	
**Insured premium (NT$)**							0.951
<18,000	469,293	89.33	93,831	89.30	375,462	89.34	
18,000–34,999	39,827	7.58	7982	7.60	31,845	7.58	
≥35,000	16,235	3.09	3258	3.10	12,977	3.09	
**DM**							0.359
Without	417,407	79.45	83,374	79.35	334,033	79.48	
With	107,948	20.55	21,697	20.65	86,251	20.52	
**HT**							<0.001
Without	410,044	78.05	81,343	77.42	328,701	78.21	
With	115,311	21.95	23,728	22.58	91,583	21.79	
**Depression**							<0.001
Without	517,915	98.58	102,600	97.65	415,315	98.82	
With	7440	1.42	2471	2.35	4969	1.18	
**Stroke**							<0.001
Without	474,272	90.28	94,073	89.53	380,199	90.46	
With	51,083	9.72	10,998	10.47	40,085	9.54	
**Dementia**							<0.001
Without	519,570	98.90	104,173	99.15	415,397	98.84	
With	5785	1.10	898	0.85	4887	1.16	
**CKD**							0.025
Without	466,555	88.81	93,106	88.61	373,449	88.86	
With	58,800	11.19	11,965	11.39	46,835	11.14	
**Season**							<0.001
Spring (March–May)	120,264	22.89	24,086	22.92	96,178	22.88	
Summer (June–August)	131,700	25.07	25,691	24.45	106,009	25.22	
Autumn (September–November)	151,076	28.76	28,862	27.47	122,214	29.08	
Winter (December–February)	122,315	23.28	26,432	25.16	95,883	22.81	
**Location**							<0.001
Northern Taiwan	213,017	40.55	45,624	43.42	167,393	39.83	
Middle Taiwan	149,055	28.37	26,980	25.68	122,075	29.05	
Southern Taiwan	131,872	25.10	27,744	26.41	104,128	24.78	
Eastern Taiwan	29,294	5.58	4402	4.19	24,892	5.92	
Outlets islands	2117	0.40	321	0.31	1796	0.43	
**Urbanization level**							<0.001
1 (The highest)	164,309	31.28	39,844	37.92	124,465	29.61	
2	236,316	44.98	48,746	46.39	187,570	44.63	
3	38,416	7.31	4060	3.86	34,356	8.17	
4 (The lowest)	86,314	16.43	12,421	11.82	73,893	17.58	
**Level of care**							<0.001
Hospital center	189,094	35.99	57,044	54.29	132,050	31.42	
Regional hospital	234,975	44.73	37,663	35.85	197,312	46.95	
Local hospital	101,286	19.28	10,364	9.86	90,922	21.63	
**Sleep disorders**							<0.001
Without	503,952	95.93	99,898	95.08	404,054	96.14	
With	21,403	4.07	5173	4.92	16,230	3.86	
**Apnea**							0.508
Without	523,931	99.73	104,776	99.72	419,155	99.73	
With	1424	0.27	295	0.28	1129	0.27	
**Insomnia**							<0.001
Without	512,702	97.59	101,590	96.69	411,112	97.82	
With	12,653	2.41	3481	3.31	9172	2.18	
**Non-apnea non-insomnia sleep disorders**							0.035
Without	517,051	98.42	103,356	98.37	413,695	98.43	
With	8304	1.58	1715	1.63	6589	1.57	

*p*-value (category variable: Chi-square/Fisher exact test; continue variable: *t*-test).

**Table 2 ijerph-19-09266-t002:** Factors of cancers by using conditional logistic regression.

Variables	Crude OR	95% CI	95% CI	*p*	Adjusted OR	95% CI	95% CI	*p*
**Sleep disorders**								
Without	Reference				Reference			
With	1.789	1.648	1.991	<0.001	1.440	1.392	1.489	<0.001
**Gender**								
Male	1.289	1.248	1.331	<0.001	1.153	1.120	1.186	<0.001
Female	Reference				Reference			
**Age group**								
18–44	Reference				Reference			
45–64	1.441	1.367	1.518	<0.001	1.442	1.367	1.520	<0.001
≥65	1.793	1.706	1.885	<0.001	1.715	1.828	1.807	<0.001
**Married**								
Without	0.816	0.345	1.672	0.492	0.842	0.452	1.726	0.374
With	Reference				Reference			
Divorce	1.208	0.897	1.397	0.124	1.113	0.571	1.453	0.288
Spouse death	1.552	0.701	2.163	0.106	1.298	0.642	1.772	0.115
Unknown	0.000	-	-	0.931	0.000	-	-	0.989
**Education**								
Elementary/junior high school	Reference				Reference			
(Vocational) high school	1.154	0.495	1.798	0.462	1.018	0.312	1.843	0.509
Univeristy/college/graduate	1.298	0.501	1.881	0.337	1.174	0.477	2.092	0.426
Others	0.000	-	-	0.872	0.000	-	-	0.911
**Insured premium (NT$)**								
<18,000	Reference				Reference			
18,000–34,999	0.935	0.832	1.052	0.264	0.952	0.847	1.071	0.415
≥35,000	0.617	0.468	0.981	0.041	0.761	0.577	1.014	0.073
**DM**								
Without	Reference				Reference			
With	1.095	1.058	1.133	<0.001	1.071	1.034	1.110	<0.001
**HT**								
Without	Reference				Reference			
With	1.948	1.818	2.177	0.001	1.914	1.884	1.945	<0.001
**Depression**								
Without	Reference				Reference			
With	3.072	2.822	3.344	<0.001	3.408	3.128	3.714	<0.001
**Stroke**								
Without	Reference				Reference			
With	1.188	1.133	1.244	<0.001	1.126	1.073	1.181	<0.001
**Dementia**								
Without	Reference				Reference			
With	1.882	1.693	2.091	<0.001	1.612	1.449	1.794	<0.001
**CKD**								
Without	Reference				Reference			
With	1.889	1.741	2.049	<0.001	1.708	1.574	1.855	<0.001
**Season**								
Spring	Reference				Reference			
Summer	1.113	1.068	1.160	<0.001	1.108	1.063	1.155	<0.001
Autumn	1.325	1.274	1.377	<0.001	1.305	1.255	1.356	<0.001
Winter	1.121	1.075	1.169	<0.001	1.111	1.062	1.157	<0.001
**Location**					**Had multicollinearity with urbanization level**			
Northern Taiwan	Reference				**Had multicollinearity with urbanization level**			
Middle Taiwan	1.340	1.296	1.386	<0.001	**Had multicollinearity with urbanization level**			
Southern Taiwan	1.189	1.147	1.233	<0.001	**Had multicollinearity with urbanization level**			
Eastern Taiwan	1.922	1.824	2.026	<0.001	**Had multicollinearity with urbanization level**			
Outlets islands	1.738	1.477	2.088	<0.001	**Had multicollinearity with urbanization level**			
**Urbanization level**								
1 (The highest)	0.673	0.647	0.700	<0.001	0.814	0.779	0.851	<0.001
2	0.725	0.699	0.752	<0.001	0.827	0.796	0.860	<0.001
3	0.833	0.786	0.881	<0.001	0.872	0.823	0.923	<0.001
4 (The lowest)	Reference				Reference			
**Level of care**								
Hospital center	0.626	0.603	0.650	<0.001	0.636	0.609	0.664	<0.001
Regional hospital	0.778	0.752	0.805	<0.001	0.774	0.747	0.801	<0.001
Local hospital	Reference				Reference			

OR: odds ratio; CI: confidence interval; Adjusted OR: adjusted variables listed in the table.

**Table 3 ijerph-19-09266-t003:** Factors of cancers stratified by variables listed in the table by using conditional logistic regression.

Sleep Disorders (With vs. Without)	With			Without			Ratio	Adjusted OR	95%CI	95%CI	*p*
Stratified	Exposure	PYs	Rate (per 10^3^ PYs)	Exposure	PYs	Rate (per 10^3^ PYs)					
**Total**	5173	288,330.74	17.94	16,230	2,377,934.41	6.83	2.629	1.440	1.392	1.489	<0.001
**Gender**											
Male	3206	166,695.38	19.23	9977	1,390,636.46	7.17	2.681	1.469	1.410	1.598	<0.001
Female	1967	121,635.36	16.17	6253	987,297.95	6.33	2.553	1.399	1.252	1.444	<0.001
**Age group**											
18–44	555	39,942.92	13.89	1318	221,566.17	5.95	2.336	1.280	1.137	1.304	<0.001
45–64	1773	97,440.27	18.20	5307	749,270.05	7.08	2.569	1.407	1.345	1.469	<0.001
≥65	2845	150,947.55	18.85	9605	1,407,098.19	6.83	2.761	1.513	1.460	1.678	<0.001
**Married**											
Without	810	38,801.75	20.88	2916	306,989.47	9.50	2.198	1.204	1.102	1.385	0.001
With	2098	144,232.45	14.55	5925	1,066,714.45	5.55	2.619	1.435	1.374	1.571	<0.001
Divorce	1144	62,067.42	18.43	4433	642,980.13	6.89	2.673	1.529	1.411	1.601	<0.001
Spouse death	1121	43,121.08	26.00	2956	360,245.12	8.21	3.168	1.671	1.520	1.798	<0.001
Unknown	0	108.04	0.00	0	1005.24	0.00	-	-	-	-	-
**Education**											
Elementary/junior high school	907	73,201.45	12.39	3373	506,601.45	6.66	1.861	1.019	0.984	1.127	0.298
(Vocational) high school	2021	119,454.24	16.92	6345	1,000,845.06	6.34	2.669	1.462	1.349	1.602	0.011
Univeristy/college/graduate	2245	94,248.57	23.82	6512	797,715.11	8.16	2.918	1.598	1.445	1.798	0.006
Others	0	1426.48	0.00	0	72,772.79	0.00	-	-	-	-	-
**Insured premium (NT$)**											
<18,000	5101	284,012.19	17.96	15,901	2,332,537.19	6.82	2.635	1.495	1.407	1.572	0.005
18,000–34,999	55	3594.56	15.30	240	37,892.06	6.33	2.416	1.302	1.215	1.480	0.010
≥35,000	17	723.99	23.48	89	7505.16	11.86	1.980	1.082	1.009	1.133	0.042
**DM**											
Without	4506	248,115.42	18.16	12,674	1,797,644.23	7.05	2.576	1.411	1.364	1.459	<0.001
With	667	40,215.32	16.59	3556	580,290.18	6.13	2.707	1.483	1.433	1.533	<0.001
**HT**											
Without	4230	241,960.51	17.48	11,192	1,570,819.26	7.12	2.454	1.344	1.299	1.390	<0.001
With	943	46,370.23	20.34	5038	807,115.15	6.24	3.258	1.785	1.725	1.845	<0.001
**Depression**											
Without	5070	286,741.71	17.68	15,819	2,348,030.54	6.74	2.624	1.338	1.290	1.402	<0.001
With	103	1589.03	64.82	411	29,903.87	13.74	4.716	2.854	2.497	2.671	<0.001
**Stroke**											
Without	4964	277,924.03	17.86	14,340	2,071,846.60	6.92	2.581	1.414	1.367	1.463	<0.001
With	209	10,406.71	20.08	1890	306,087.81	6.17	3.252	1.782	1.722	1.897	<0.001
**Dementia**											
Without	5146	287,148.59	17.92	15,871	2,333,477.43	6.80	2.635	1.443	1.382	1.487	<0.001
With	27	1182.15	22.84	359	44,456.98	8.08	2.828	1.519	1.498	1.601	<0.001
**CKD**											
Without	4986	281,046.24	17.74	15,758	2,322,854.16	6.78	2.615	1.433	1.372	1.479	<0.001
With	187	7284.50	25.67	472	55,080.25	8.57	2.996	1.641	1.586	1.708	<0.001
**Season**											
Spring	997	62,060.76	16.06	3489	516,204.92	6.76	2.377	1.302	1.259	1.346	<0.001
Summer	1242	70,358.21	17.65	3961	568,305.95	6.97	2.533	1.387	1.341	1.435	<0.001
Autumn	1699	84,211.27	20.18	4930	728,515.28	6.77	2.981	1.633	1.579	1.689	<0.001
Winter	1235	71,700.50	17.22	3850	564,908.26	6.82	2.527	1.384	1.338	1.432	<0.001
**Urbanization level**											
1 (The highest)	1720	101,539.67	16.94	4420	661,751.90	6.68	2.536	1.389	1.343	1.469	<0.001
2	2370	135,786.88	17.45	6921	1,069,017.83	6.47	2.696	1.477	1.428	1.571	<0.001
3	190	12,557.40	15.13	1100	198,584.58	5.54	2.732	1.496	1.411	1.598	<0.001
4 (The lowest)	893	38,446.79	23.23	3789	448,580.10	8.45	2.750	1.506	1.402	1.672	<0.001
**Level of care**											
Hospital center	2153	147,056.04	14.64	3898	709,805.57	5.49	2.666	1.420	1.305	1.477	<0.001
Regional hospital	2161	112,013.98	19.29	7564	1,128,912.29	6.70	2.879	1.557	1.425	1.701	<0.001
Local hospital	859	29,260.72	29.36	4768	539,216.55	8.84	3.320	1.820	1.558	1.978	<0.001

PYs = person-years; Adjusted OR = adjusted odds ratio: adjusted for the variables listed in Table 2; CI = confidence interval.

## Data Availability

Data are available from the NHIRD published by the Taiwan NHI Administration. Data cannot be made publicly available due to legal constraints placed by the Taiwanese government in regard to the “Personal Information Protection Act.” Data requests can be submitted to the NHIRD as formal proposals.

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
