# Peer review of "The Combination of Sleep Disorders and Depression Significantly Increases Cancer Risk: A Nationwide Large-Scale Population-Based Study"

_ijerph, 2022, doi:10.3390/ijerph19159266_

Round 1

Reviewer 1 Report

This paper is very interesting and potentially clinically relevant. The last sentence of the paper "a meticulous cancer risk evaluation is recommended for patients with both sleep disorders and depression" is the real take home message of this paper. This concept could be stated more extensively by the Authors, both in the Discussion and in the Conclusions.

I think the title of the paper could mirror this take-home message, and become more interesting if changed like : "The combination of sleep disorders and depression significantly increases cancer risk: a nationwide large-scale population-based study".

Each study based on a half million people population should be considered for publication, given its statistical strength. However, I think the Authors should explain why the incidence of a high-prevalence tumor in the general aged male population, such as prostate cancer, is not so frequent in the observed patients. Is prostate cancer less common than esophagus cancer in Taiwan? Is the low incidence of prostate cancer based on national screening programs, such as the breast cancer screening program that the Authors mention? I think a comment on the low incidence of prostate cancer should be made by the Authors. Is it possible that sleep disorders/depression show their influence more on certain types of cancers, such as digestive tract cancers, and not on others? And which could be the mechanism?

If the low incidence of prostate cancer  in the observed population is due to the fact that this tumor is probably not linked to sleep disorders/depression status how can this be explained? Can we drain any conclusion to recommend specific urological screenings in the general aged male population? Which cancers should we be more concerned with?

Is prostate cancer less frequent in Taiwan as compared to Europe/US? In these countries, a personalised approach to the very frequent urological cancers is currently being developed, based on genetics, individualised cellular differentiation patterns and patients specific biomarkers (well reviewed in Mancini M., et al, Urologia 2016. Stem cell, biomarkers and genetic profiling: approaching future challenges in Urology). We could propose to include also sleeping patterns and mental status in the individualised approach to urological cancers. Can the Authors speculate on this point? Suggest a strategy? This would make the discussion/conclusion very updated and clinically relevant from a practical point of view.

I think the paper could be significantly improved with the minor changes proposed.

Author Response

Comments/ questions from Reviewer 1

This paper is very interesting and potentially clinically relevant. The last sentence of the paper "a meticulous cancer risk evaluation is recommended for patients with both sleep disorders and depression" is the real take home message of this paper. This concept could be stated more extensively by the Authors, both in the Discussion and in the Conclusions.

I think the title of the paper could mirror this take-home message, and become more interesting if changed like : "The combination of sleep disorders and depression significantly increases cancer risk: a nationwide large-scale population-based study".

Response: Thanks for your suggestion, we had changed the title of the paper to "The combination of sleep disorders and depression significantly increases cancer risk: a nationwide large-scale population-based study". The concept of our paper "a meticulous cancer risk evaluation is recommended for patients with both sleep disorders and depression" has been stated more extensively in conclusions.

Each study based on a half million people population should be considered for publication, given its statistical strength. However, I think the Authors should explain why the incidence of a high-prevalence tumor in the general aged male population, such as prostate cancer, is not so frequent in the observed patients. Is prostate cancer less common than esophagus cancer in Taiwan? Is the low incidence of prostate cancer based on national screening programs, such as the breast cancer screening program that the Authors mention? I think a comment on the low incidence of prostate cancer should be made by the Authors. Is it possible that sleep disorders/depression show their influence more on certain types of cancers, such as digestive tract cancers, and not on others? And which could be the mechanism?

If the low incidence of prostate cancer in the observed population is due to the fact that this tumor is probably not linked to sleep disorders/depression status how can this be explained? Can we drain any conclusion to recommend specific urological screenings in the general aged male population? Which cancers should we be more concerned with?

Is prostate cancer less frequent in Taiwan as compared to Europe/US? In these countries, a personalised approach to the very frequent urological cancers is currently being developed, based on genetics, individualised cellular differentiation patterns and patients specific biomarkers (well reviewed in Mancini M., et al, Urologia 2016. Stem cell, biomarkers and genetic profiling: approaching future challenges in Urology). We could propose to include also sleeping patterns and mental status in the individualised approach to urological cancers. Can the Authors speculate on this point? Suggest a strategy? This would make the discussion/conclusion very updated and clinically relevant from a practical point of view.

I think the paper could be significantly improved with the minor changes proposed.

Response: As shown in figure 3, the subgroup analysis of specific cancer types, our data suggest an increased risk of 12 types of cancer in patients with both sleep disorders and depression, including prostate cancer. In other words, patients with both sleep disorders and depression associated with a higher risk of prostate cancer than general population.

Esophageal and hematologic cancers are the two specific types of cancer showed extremely high risk in patients with both sleep disorder and depression (Figure 3). The adjusted OR of prostate cancer was not as high as esophageal cancer or hematologic cancer. However, we don’t really understand the true biological mechanisms that why sleep disorders and depression affects esophageal cancer and hematologic cancer more. Thanks for the idea you shared, we would like to investigate further basic mechanism of this phenomenon in our future studies.

Reviewer 2 Report

Dear authors,

I read with great interest your paper on the increased risk to develop cancer in patients with sleep disorders and depression.

Most studies investigate cancer induced sleep disorders and depression, therefore sleep disorders and depression induced cancer sounds an interesting novelty. 

The cohort retrospective study is well structured and well conducted.

Please clarify in figure 1 (and also write it in the methods : "the last sleep disorders diagnosis before the first cancer desease diagnosis <2 years in retrospective duration". I can immagine what it means but the sentence is unclear.

In my opinion the paper is absolutely worth publication.

Congratulations and kind regards,

the reviewer. 

Author Response

Dear authors,

I read with great interest your paper on the increased risk to develop cancer in patients with sleep disorders and depression.

Most studies investigate cancer induced sleep disorders and depression, therefore sleep disorders and depression induced cancer sounds an interesting novelty. 

The cohort retrospective study is well structured and well conducted.

Please clarify in figure 1 (and also write it in the methods : "the last sleep disorders diagnosis before the first cancer desease diagnosis <2 years in retrospective duration". I can immagine what it means but the sentence is unclear.

In my opinion the paper is absolutely worth publication.

Congratulations and kind regards,

the reviewer. 

Response: Thanks for your suggestion, we had made the clarification of the exclusion criteria as we marked in the manuscript.

Reviewer 3 Report

This article reports the results of a study who sought to assess the interaction effect between sleep disorders and cancer presence in a large general population in Taiwan (n = 105,071 with cancer vs. 420,284 without cancer). It covers a very interesting topic in health psychology, i.e. the interaction between psychological and physical conditions. However, a methodological limitation precludes sustaining the main finding. Modestly, I present below some suggestions to improve the manuscript:

COMMENT 1. The introduction seems very short to me. As far as I understand, it briefly covers the following points: (1) sleep and sleep disorders, (2) sleep disorders in patients with cancer, (3) sleep disorders and mental disorders (e.g., major depression disorder), (4) cancer-induced depression vs depression as a risk factor of cancer. I suggest explaining in deep these ideas, specially the central one, which authors stated in page 2: “Most researchers have focused on cancer-induced depression when investigating the relationship between depression and cancer, and only few have explored depression as a risk factor for subsequent cancer.” Alternatively, as a free suggestion, I propose to develop the following ideas in this order of appearance: (1) cancer (an overview of the disease), (2) psychological comorbidities (among them, sleep disorders and depression), (3) sleep disorders and depression in cancer patients, (4) interaction between psychological factors and cancer (what is known, from which perspective it has been studied, biomedical vs. biopsychosocial model), (5) objective.

COMMENT 2. Please, kindly update your references, try to include as many recent publications (< 5 years) as possible. Keep those old articles that are very relevant, but recent research should be the most frequent throughout the paper.

COMMENT 3. Study subjects subsection (lines 80-96). How was depression assessed or diagnosed? It is considered as a covariate; however, the title of the paper and the introduction indicates that this was studied at the same level of importance than sleep disorders. So, should not be that explain in the same detail than sleep disorder? For a suspicious reader, and as is shown in the paper, it looks like authors gave importance to this variable in the introduction section and included it in the objective of the study when after analysing data it showed significance. For instance, in Line 103 author say, “Covariates that were potential confounders in the association between sleep disorders and cancer included age, sex, and underlying chronic diseases (à here appears depression).”

COMMENT 4. Please, mention again the covariates in the Statistical Analysis subsection (lines 114-115). Although authors have previously mentioned them, they have to appear here.

COMMENT 5. Please, consider relocating the first paragraph of the Results section as an independent subsection named Participants in the Methods section. However, check the journals requirements after doing that.

COMMENT 6. Where are Table 1, 2, and 3? There are no tables in the manuscript.

COMMENT 7. Do authors assume causation in the interpretation of their results? It seems that: Line 141-142. “A significantly higher risk of cancer development was observed in patients with sleep disorders.” I might be wrong, but this analysis does not allow seeing causation, but correlation between variables. Take into consideration that you have data from one moment. Thus, considering your results, it could be said both “that patients with coexisting sleep disorders and depression were 6.85 times more likely to develop cancer than the general population (line 180-181)” and that patients with cancer are 6.85 times more likely to develop sleep disorders and depression than the general population. This issue is central and affects all the paper (from title to discussion). You have to solve this.

Author Response

COMMENT 1. The introduction seems very short to me. As far as I understand, it briefly covers the following points: (1) sleep and sleep disorders, (2) sleep disorders in patients with cancer, (3) sleep disorders and mental disorders (e.g., major depression disorder), (4) cancer-induced depression vs depression as a risk factor of cancer. I suggest explaining in deep these ideas, specially the central one, which authors stated in page 2: “Most researchers have focused on cancer-induced depression when investigating the relationship between depression and cancer, and only few have explored depression as a risk factor for subsequent cancer.” Alternatively, as a free suggestion, I propose to develop the following ideas in this order of appearance: (1) cancer (an overview of the disease), (2) psychological comorbidities (among them, sleep disorders and depression), (3) sleep disorders and depression in cancer patients, (4) interaction between psychological factors and cancer (what is known, from which perspective it has been studied, biomedical vs. biopsychosocial model), (5) objective.

 Response: Thanks for your suggestion, we had increased the length of introduction and described more about sleep disorders and depression in cancer patients, interaction between psychological factors and cancer.

Page 2, line 9-15: When investigating the relationship between depression and cancer, most re-searchers have focused on cancer-induced depression, and only few have explored depression as a risk factor for subsequent cancer. Cancer is caused by a variety of variables, including genetics, poor behaviors, the environment, and psychosocial factors including depression. Since the 1980s, research has shown that the immune system, the endocrine system, cancer metastasis, treatment tolerance, and other processes are all impacted by depression.

COMMENT 2. Please, kindly update your references, try to include as many recent publications (< 5 years) as possible. Keep those old articles that are very relevant, but recent research should be the most frequent throughout the paper.

 Response: Thanks for your suggestion, we had enrolled many recent publication related to our studies.

Page 2, line 4-6: However, several studies have shown that sleep disorders may be associated with the development of cancers, such as breast cancer [9], colorectal cancer [10], lung cancer [11, 12], and liver cancer [12].

COMMENT 3. Study subjects subsection (lines 80-96). How was depression assessed or diagnosed? It is considered as a covariate; however, the title of the paper and the introduction indicates that this was studied at the same level of importance than sleep disorders. So, should not be that explain in the same detail than sleep disorder? For a suspicious reader, and as is shown in the paper, it looks like authors gave importance to this variable in the introduction section and included it in the objective of the study when after analysing data it showed significance. For instance, in Line 103 author say, “Covariates that were potential confounders in the association between sleep disorders and cancer included age, sex, and underlying chronic diseases (à here appears depression).”

Response: Thanks for your suggestion, we had made the correction.

Page 3, line 11-14: The diagnoses of all depression were made by certified psychiatrists and in accordance with the DSM-V criteria, which required at least two outpatient visits or one admission record for ICD-9-CM codes: 296.2-296.3, 296.82, 300.4, 311.

COMMENT 4. Please, mention again the covariates in the Statistical Analysis subsection (lines 114-115). Although authors have previously mentioned them, they have to appear here.

 Response: Thanks for your suggestion, we had made the correction.

COMMENT 5. Please, consider relocating the first paragraph of the Results section as an independent subsection named Participants in the Methods section. However, check the journals requirements after doing that.

Response: Thanks for your suggestion; however, we decided to leave the first paragraph of the Results to the original place. Because it included the results of patient characteristics.

COMMENT 6. Where are Table 1, 2, and 3? There are no tables in the manuscript.

Response: They were attached separately, in different file.

COMMENT 7. Do authors assume causation in the interpretation of their results? It seems that: Line 141-142. “A significantly higher risk of cancer development was observed in patients with sleep disorders.” I might be wrong, but this analysis does not allow seeing causation, but correlation between variables. Take into consideration that you have data from one moment. Thus, considering your results, it could be said both “that patients with coexisting sleep disorders and depression were 6.85 times more likely to develop cancer than the general population (line 180-181)” and that patients with cancer are 6.85 times more likely to develop sleep disorders and depression than the general population. This issue is central and affects all the paper (from title to discussion). You have to solve this.

 Response: Thanks for your suggestion, we had discussed about the description we made “A significantly higher risk of cancer development was observed in patients with sleep disorders.”, however, we thought it is an objectively stated facts lack of causation.

Round 2

Reviewer 3 Report

Thank you for answering my comments. After these corrections the paper has improved, however there are some points to selve yet:

Introduction: 

Comment 1. In general terms, the introduction could be improved by developing in deep the present ideas. 

Comment 2. Page 2, line 9-15: "When investigating the relationship between depression and cancer, most re-searchers have focused on cancer-induced depression, and only few have explored depression as a risk factor for subsequent cancer. Cancer is caused by a variety of variables, including genetics, poor behaviors, the environment, and psychosocial factors including depression. Since the 1980s, research has shown that the immune system, the endocrine system, cancer metastasis, treatment tolerance, and other processes are all impacted by depression." Please, provide citations for this information. Where all these ideas are from?

Comment 3. Finally, I have understood that longitudinal data was collected. I think this is not clearly described. Authors should explicitly mention in the methods section that the previous medical history of patients that developed a cancer was gathered and that this information was studied as the 'predictor'. This idea should be stated also in the abstract.

Author Response

Comments/ questions from Reviewer 3

Comment 1. In general terms, the introduction could be improved by developing in deep the present ideas.

Response: Thanks for your suggestion, we had expanded and developed the introduction in deep according to the presented ideas.

Page 2, line 9

Since the 1980s, research has shown that the immune system, the endocrine system, cancer metastasis, treatment tolerance, and other processes are all impacted by depression [46]. It has been suggested that depression and anxiety, the two most prevalent mental disorders with past-year prevalence rates of 5% and 7%, respectively, in the general population, may potentially play an etiologic role and have a prognostic impact on cardiovascular diseases, such as stroke and coronary heart disease [47, 48]. Furthermore, depression is much more prevalent among cancer patients, affecting up to 20% of cancer patients [49]. However, when investigating the relationship between depression and cancer, most researchers have focused on cancer-induced depression, and only few have explored depression as a risk factor for subsequent cancer. Although some meta-analysis studies have investigated the relationship between depression and overall cancer risk over the past decade [14, 15], the results of these studies are inconsistent or require larger sample sizes for confirmation.

Comment 2. Page 2, line 9-15: "When investigating the relationship between depression and cancer, most re-searchers have focused on cancer-induced depression, and only few have explored depression as a risk factor for subsequent cancer. Cancer is caused by a variety of variables, including genetics, poor behaviors, the environment, and psychosocial factors including depression. Since the 1980s, research has shown that the immune system, the endocrine system, cancer metastasis, treatment tolerance, and other processes are all impacted by depression." Please, provide citations for this information. Where all these ideas are from?

Response: Thanks for your suggestion, we had cited the references.

Comment 3. Finally, I have understood that longitudinal data was collected. I think this is not clearly described. Authors should explicitly mention in the methods section that the previous medical history of patients that developed a cancer was gathered and that this information was studied as the 'predictor'. This idea should be stated also in the abstract.

Response: Thanks for your suggestion, we had stated this description in the method and abstract.

Page 3, line 13: The chronic diseases of patients that may developed cancer were gathered and studied as the predictor, included hypertension (HTN, ICD-9-CM codes: 401, 404, and 405), diabetes mellitus (DM) (ICD-9-CM code: 250), stroke (ICD-9-CM code: 430-438), dementia (ICD-9-CM codes: 290, 294.1, and 331.0), chronic kidney disease (CKD, ICD-9-CM code: 585), and depression.
